# Updated meta-analysis of fractional flow reserve versus coronary angiography for guiding percutaneous coronary intervention

**Fahui Yin**[ID][�]*, **Yong Zhang**[�], **Xueqian Zhang, Yangang Chen, Xuelian Cui**

Department of Cardiovascular Disease, Liangzhou Hospital, Wuwei, Gansu, China

উ These authors contributed equally to this work.
* 414380580@qq.com

**Data availability statement:** All relevant data are within the manuscript and its Supporting information files.

## Abstract

**Background:** Fractional Flow Reserve (FFR) has been widely utilized in clinical practice for decades,however, the comparative clinical outcomes of FFR-guided versus coronary angiography (CAG)-guided percutaneous coronary intervention (PCI) still warrant further evaluation.

**Methods and materials:** Randomized controlled trials (RCTs) comparing FFR-guided and CAG-guided PCI were systematically searched in PubMed, Embase and the Cochrane library databases from their respective inception to December 31, 2023. Primary endpoints included the incidence of major adverse cardiovascular events (MACE), all cause mortality, myocardial infarction (MI) and target vessel revascularization(TVR). Stratified analyses were performed to evaluate the effects of FFR-guided versus CAG-guided PCI across different follow-up periods (short-term and long-term) and patient cohorts (acute coronary syndrome (ACS) and non-ACS patients).

**Results:** This meta-analysis included eight RCTs involving 4,433 patients, with four studies reporting 1-year outcomes and four reporting outcomes beyond one year. Among these, 5 studies focused on non-ACS patients, and three included ACS patients, with a significant male predominance (3,437 vs. 996 females). By follow-up duration, FFR-guided PCI demonstrated significant long-term reductions in MACE (OR: 0.76, 95% CI: 0.60-0.96, P = 0.022) and MI (OR: 0.65, 95% CI: 0.45-0.93, P = 0.018), but no significant short-term benefits were observed for MACE (OR: 0.85, 95% CI: 0.67-1.08, P = 0.194), MI (OR: 0.85, 95% CI: 0.63-1.16, P = 0.307), or all-cause mortality (short-term: OR: 0.77, 95% CI: 0.47-1.26, P = 0.296; long-term: OR: 0.74, 95% CI: 0.50-1.09, P = 0.123). By patient type, FFR-guided PCI significantly reduced MACE (OR: 0.82, 95% CI: 0.68-0.99, P = 0.038), MI (OR: 0.76, 95% CI: 0.58-0.99, P = 0.039), and TVR (OR: 0.78, 95% CI: 0.61-0.99, P = 0.036) in non-ACS patients, but no significant differences were observed in ACS patients for MACE (OR: 0.76, 95% CI: 0.53-1.08, P = 0.127), all-cause mortality (OR: 0.60, 95% CI: 0.35-1.02, P = 0.060), MI (OR: 0.77, 95% CI: 0.47-1.25, P = 0.294), or TVR (OR: 0.98, 95% CI: 0.48-2.02, P = 0.315). Sensitivity analysis confirmed the robustness of these findings.

**Funding:** The author(s) received no specific funding for this work.

**Competing interests:** No authors have competing interests.

**Conclusions:** FFR-guided PCI is superior to CAG-guided PCI in reducing MACE and MI in long-term and non-ACS patients but shows no advantage in short-term or ACS populations. FFR should be avoided in patients presenting with ACS in routine clinical practice.

## Introduction

Coronary artery disease (CAD) remains the leading cause of death worldwide, imposing a substantial global health burden on humanity [1]. As the "golden standard" for CAD diagnosis, coronary angiography (CAG) has long served as the primary guidance method for percutaneous coronary intervention (PCI). However, due to its inherent limitations in accuracy, it usually overestimates or underestimates the severity of the intermediate coronary stenosis and cannot determine the hemodynamic significance of coronary stenosis, the clinical value of anatomical stenosis assessment based on CAG has been challenged by FFR-a technique that enables scientific measurement of inducible ischemia in target vessels via a specialized pressure guidewire [2].

FFR is defined as the ratio of the pressure distal to stenosis and the aortic pressure in a condition of maximum myocardium hyperemia. Using a specialized coronary pressure wire, FFR could provide a readily available, straightforward and quantitative technique to evaluate the physiologic significance of a coronary lesion. When the FFR value is greater than 0.8, the stenosis is considered non-significant, whereas a value less than 0.75 indicates coronary ischemia [3]. The European Society of Cardiology has recommended FFR as a Class IA indication for guiding percutaneous coronary revascularization, while the American Heart Association practice guidelines have classified it as Class IIA. [4,5].

Many studies have investigated the clinical outcomes of FFR-guided versus CAG-guided PCI in patients with CAD. The Fraction Flow Reserve Versus Angiography for Multivessel Evaluation (FAME) study showed FFR-guided PCI had lower rates of mortality, reinfarction and MACEs compared with standard angiography-guided PCI [6]. However, the DEFER-DES trail (Fractional Flow Reserve–Guided Versus Routine Drug-Eluting Stent Implantation in Patients With Intermediate Coronary Stenosis) demonstrated that no significant differences were observed in cardiac death, revascularization and MACEs between the two groups after five years of follow up [7]. Additionally, although previous meta-analyses have compared these two strategies, most included observational studies and enrolled patients who underwent different revascularization procedures (some patients underwent PCI, while others opted for coronary artery bypass graft (CABG)) [8,9]. In summary, previous studies presented mixed findings regarding the clinical outcomes of FFR-guided versus CAG-guided PCI in patients with CAD. Therefore, the aim of this study is to further investigate the clinical outcomes of these two PCI guidance strategies.

## Materials and methods

The Preferred Reporting Items for Systematic Reviews and Meta-Analyses (PRISMA) were used to guide the reporting of this meta-analysis. Elements of this quantitative method systematic literature review include data sources and search strategy, inclusion and exclusion criteria, subgroup analysis, data collection and extraction, risk of bias assessment, and statistical analysis.

## Data sources and search strategy

PubMed, Embase and the Cochrane library data bases were searched to identify articles comparing CAG and FFR-guided PCI before December 31,2023. All articles were restricted to English. According to the "PICOS" principle of evidence-based medicine, the following medical subject headings and search terms were used: (1) fractional flow reserve; (2) coronary angiography; (3) percutaneous coronary intervention. Additionally, references of relevant articles and conference reports were screened to uncover potential suitable studies. A comprehensive search strategy was developed after examination of titles and abstracts of related journal articles by using the search terms (Supplementary files).

## Inclusion and exclusion criteria

The PICO (Population, Intervention, Comparison, and Outcome) framework was used to formulate the research question. Studies were eligible for inclusion included if they met the following criteria: (1) Randomised controlled trails (RCTs) ; (2) Comparing FFR-guided and angiography-guided PCI; (3) Reporting at least one of the following outcomes, including MACEs, all-cause mortality, MI, and TVR. Studies were excluded if they were: (1) Non-RCTs; (2) Studies included the patients underwent CABG; (3) Did not report relevant outcomes; (4) Did not allow for the extraction of detailed data.

## Subgroup analysis

In this meta-analysis, subgroup analysis were conducted based on the follow-up period and patient cohorts. Regarding the follow-up period, the studies were divided into short-term subgroup (≤1 year) and long-term follow-up subgroup (>1 year). In terms of patient cohorts, the studies were divided into acute coronary syndrome (ACS) subgroup and non-ACS subgroup. In the ACS subgroup, we included studies involving both STEMI and NSTEMI patients, while the non-ACS subgroup comprised studies of CAD patients with multivessel disease, intermediate coronary stenosis, or true coronary bifurcation lesions. This type of subgroup analysis not only detect the potential heterogeneity among different studies, but also allow for a better comparison of the outcomes of FFR-guided and CAG-guided PCI in different patient populations and clinical backgrounds.

## Data collection and extraction

Firstly, all records were imported into the Endnote software (version X7, Thomson Reuters, U.S) to perform automatic duplication removal during the initial search. After eliminating duplicates, three reviewers (Yin FH, Zhang Y, and Zhang XQ) independently screened the titles and abstracts and excluded the irrelevant records. Subsequently, the full-text articles of the remaining records were retrieved for in-depth screening. Additionally, the reference of relevant articles and conference reports were also screened for potential suitable studies. A standard data extraction form was designed prior to data extraction. From each included study, the following information was extracted: the first author, year of publication, nationality, follow-up duration, sample size of each study, baseline characteristics, and the clinical outcomes (Table 1). Any discrepancy in data collection and extraction were resolved through consensus.

**Table 1. Baseline characteristics of the included studies.**

| Study | Quintella 2019 | | Puymirat 2021 | | Nunen 2015 | | Park 2015 | |
|---|---|---|---|---|---|---|---|---|
| Follow-up (year) | 1.5 | | 1 | | 5 | | 5 | |
| Nationality* | Brazil | | France | | Netherland | | Korea | |
| Type of patients | MVD | | STEMI and MVD | | MVD | | ICS | |
| Group | CAG | FFR | CAG | FFR | CAG | FFR | CAG | FFR |
| Number of patients | 35 | 34 | 577 | 586 | 429 | 436 | 115 | 114 |
| Age, mean (SD), y | 60± 9 | 63 ± 8 | 62 ± 11 | 63 ± 11 | 64± 10 | 65 ± 10 | 63 ± 10 | 62± 10 |
| Male patients, No.(%) | 22 (63) | 25 (74) | 468 (81) | 498 (85) | 318 (74) | 328 (75) | 83(73) | 87(76) |
| Diabetes Mellitus,No.(%) | 12 (34) | 12 (35) | 82(14) | 107(18) | 107 (25) | 98 (22) | 39 (34) | 30 (26) |
| Hypertension, No.(%) | 26 (74) | 25 (74) | 262(45) | 253(43) | 277 (65) | 259 (59) | 73 (64) | 65 (57) |
| Dyslipedemia, No.(%) | 26 (74) | 24 (71) | 237(41) | 232(39) | 316 (74) | 307 (70) | 78 (68) | 80 (70) |
| Current smoker, No.(%) | 9 (26) | 10 (29) | 210(36) | 235(40) | 130 (30) | 111 (25) | 38 (33) | 30 (26) |
| Previous MI, No.(%) | 7 (20) | 8 (24) | 31(5) | 45(8) | 155 (36) | 154 (35) | 20 (17) | 22 (19) |
| Outcomes† | 1, 2, 4 | | 1, 2, 3, 4 | | 1, 2, 3, 4 | | 1, 4 | |
| Study | Chen 2015 | | Zhang 2016 | | Tonino6 2009 | | Lee 2023 | |
| Follow-up (year) | 1 | | 1 | | 1 | | 3.5 | |
| Nationality* | China | | China | | Netherland | | Korea | |
| Type of patients | CAD with TCBL | | NSTEMI | | MVD | | STEMI and MVD | |
| Group | CAG | FFR | CAG | FFR | CAG | FFR | CAG | FFR |
| Number of patients | 160 | 160 | 110 | 110 | 496 | 509 | 278 | 284 |
| Age, mean (SD),y | 65 ± 9 | 65 ± 9 | 70 ± 3 | 70 ± 4 | 64 ± 10 | 65 ± 10 | 63 ± 12 | 64 ± 11 |
| Male patients, No.(%) | 116 (73) | 121 (76) | 78 (71) | 75 (68) | 360 (73) | 384 (75) | 234(84) | 240(85) |
| Diabetes Mellitus,No.(%) | 43 (27) | 48 (30) | 36 (33) | 40 (36) | 125 (25) | 123 (24) | 86(31) | 97(34) |
| Hypertension, No.(%) | 106 (68) | 116 (73) | 83 (76) | 81 (74) | 327 (66) | 312 (61) | 152(55) | 151(53) |
| Dyslipedemia, No.(%) | 32 (20) | 27 (17) | 93 (85) | 90 (82) | 362 (73) | 366 (72) | 107(39) | 121(43) |
| Current smoker, No.(%) | 64 (40) | 66 (41) | 31 (28) | 29 (26) | 156 (32) | 138 (27) | 105(38) | 91(32) |
| Previous MI, No.(%) | 19 (12) | 12 (8) | 24 (22) | 23 (21) | 180 (36) | 187 (37) | 7(3) | 7(3) |
| Outcomes† | 1, 2, 3, 4 | | 1, 2, 3 | | 1, 2, 3, 4 | | 1, 2, 3, 4 | |

Values are presented as percentage; "*" indicates the first author's nationality; "†" four outcomes were included in this meta-analysis:

1 indicates MACE;
2 indicates all cause mortality;
3 indicates myocardial infarction;
4 targeted vessel revascularization.
MVD = multi-vessel disease; STEMI = ST segment elevation myocardial infarction; ICS = intermediate coronary stenosis; CAD = coronary artery disease; TCBL = true coronary bifurcation lesion.

## Risk of bias assessment

The methodological quality of included randomized controlled trials was rigorously evaluated using the Revised Cochrane Risk of Bias tool (ROB 2.0) [10]. This comprehensive assessment examined five critical domains: (1) randomization process, (2) deviations from intended interventions, (3) missing outcome data, (4) measurement of outcomes, and (5) selection of

reported results. Two independent reviewers conducted the evaluations using standardized criteria, with any discrepancies resolved through discussion until consensus was achieved.

## Statistical analysis

Continuous variables were expressed as mean ± standard deviation (SD), while categorical variables were presented as numbers and percentages. Statistical analyses were performed using Stata software (version 18.0, Stata Corp, College Station, Texas). Heterogeneity among the included studies was assessed using the Chi-square test and quantified by the $I^2$ statistic. Given that all outcomes in this study were dichotomous data, they were pooled and expressed as odds ratios (ORs). A fixed-effect model (Mantel-Haenszel method) was applied when $I^2 < 50\%$, indicating low heterogeneity, whereas a random-effect model (DerSimonian-Laird method) was used for $I^2 \geq 50\%$, indicating significant heterogeneity, with greater weight assigned to smaller studies. Subgroup analyses were conducted based on follow-up duration and patient cohorts to explore potential sources of heterogeneity and to compare fractional flow reserve (FFR) and coronary angiography (CAG) under different clinical scenarios. Sensitivity analysis was performed by comparing fixed- and random-effect models. Due to the limited number of included studies (fewer than ten), funnel plots were not utilized, as their power to detect asymmetry would be insufficient. Results were reported as ORs with 95% confidence intervals (CIs), and a p-value < 0.05 was considered statistically significant.

# Results

## Searching results and quality assessment

The initial literature searching yielded 1959 articles. An additional 9 articles were identified through manual reference checking of the included studies. Duplicates were removed through automated and manual deduplication processes. After screening titles and abstracts, 29 articles were retained by excluding irrelevant ones (as detailed in S1 Table). Ultimately, 8 studies met inclusion criteria and were enrolled into this meta-analysis [6,11–17]. The study selection process is illustrated in Fig 1. The revised Cochrane risk of bias (Rob2) tool was utilized to assess the quality of enrolled RCTs. Most studies described the detailed process of random sequence generation. However, since blinding of participants and intervention providers was not feasible, potential deviations from the intended interventions due to the experimental context were likely, though difficult to ascertain from the articles. Six studies employed independent clinical data and events committees to evaluate endpoints in a blinded manner. All studies provided detailed clinical endpoint events and accounted for missing data, but the source of these data (whether from multiple eligible outcome measurements) was unclear. The results of the risk of bias assessment are presented in S1 Fig and S2 Fig.

## Baseline characteristics of the included studies

Eight studies involving 4,433 patients that met our predefined inclusion criteria were included into this meta-analysis [6,11–17]. Among these, three studies compared FFR-guided and CAG-guided PCI in patients with multivessel patients [6,14,15] two enrolled patients with STEMI and multivessel patients [13,17], one focused on with older patients with NSTEMI [11], and the other two recruited patients with intermediate coronary stenosis and true coronary bifurcation lesions, respectively [7,12]. Baseline characteristics of the studies are presented in Table 1. The follow up duration was ranged from 1 year to 5 years, and the mean age ranged from 59.5 to 70 years. Male patients were significantly outnumbered female patients in overall population (3437 versus 996, respectively), with 24.5% having diabetes, 57.9%

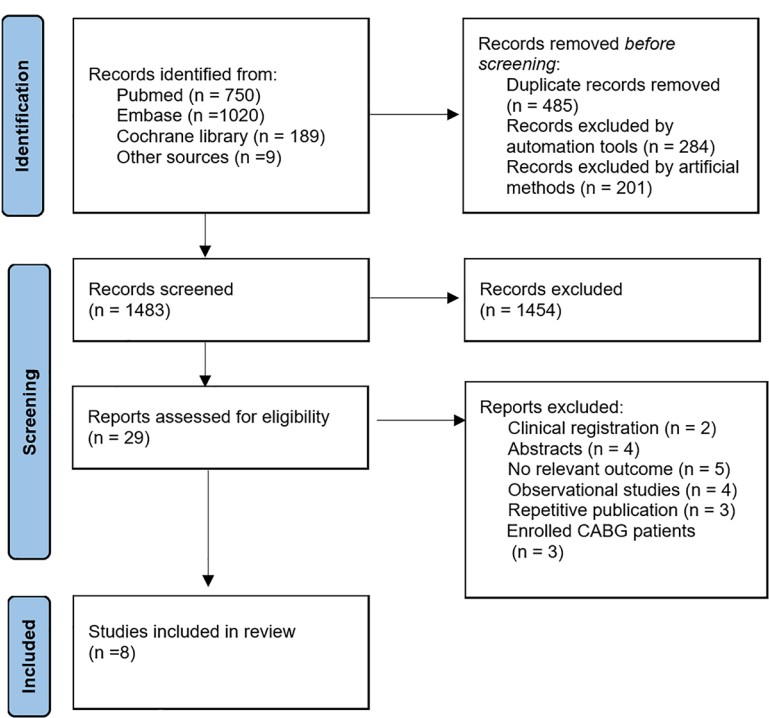

**Fig 1. Flow Diagram of studies selection process.**

having hypertension, 56.3% having dyslipidemia, and 32.8% being current smokers. No significant difference was observed between the two groups. Based on follow-up duration, the studies were further divided into short-term (≤1 year) and long -term (>1 year) follow-up groups: four studies with one-year follow-up belonged to the short-term subgroup [6,11–13], while the other four belonged to long-term subgroup [14–17]. Additionally, based on patient cohorts, the studies were divided into ACS [11,13,17] and non-ACS subgroups [6,12,14–16].

## Clinical outcomes of FFR-guided and CAG-guided PCI

**MACE.** A total of 8 studies involving 4433 patients reported MACE rates, with 4 studies providing 1-year outcomes [6,11–13] and the remaining 4 reporting outcomes beyond 1 year [14–17]. Stratified analysis based on follow up duration revealed no significant difference in short-term outcomes between the FFR and CAG groups (OR: 0.85, 95%CI 0.67-1.08, P = 0.194, $I^2$ = 35.9%). Sensitivity analysis confirmed the robustness of the results, as the pooled effect sizes calculated using both fixed-effect and random-effects models remained consistent. However, FFR-guided PCI demonstrated significant long-term benefits in reducing MACE rates compared to CAG-guided PCI (OR:0.76, 95%CI 0.60- 0.96, P = 0.022, $I^2$ = 47.6%) (Fig 2A). Among the included studies, 5 focused on non-ACS patients [6,12,14–16], while 3 included ACS patients [11,13,17]. Subgroup analysis by patient cohorts showed that FFR-guided PCI was associated with greater reductions in MACE rates in the non-ACS subgroup (OR:0.82, 95%CI 0.68- 0.99, P = 0.038, $I^2$ = 0.0%). no significant difference was observed in the ACS subgroup (OR:0.76, 95%CI 0.53- 1.08, P = 0.127, $I^2$ = 76.9%) (Fig 3A).

**All-cause mortality.** When assessing all-cause mortality, 7 studies involving 4204 patients reported relevant data [6,11–15,17], including 4 studies with 1-year outcomes [6,11–13] and

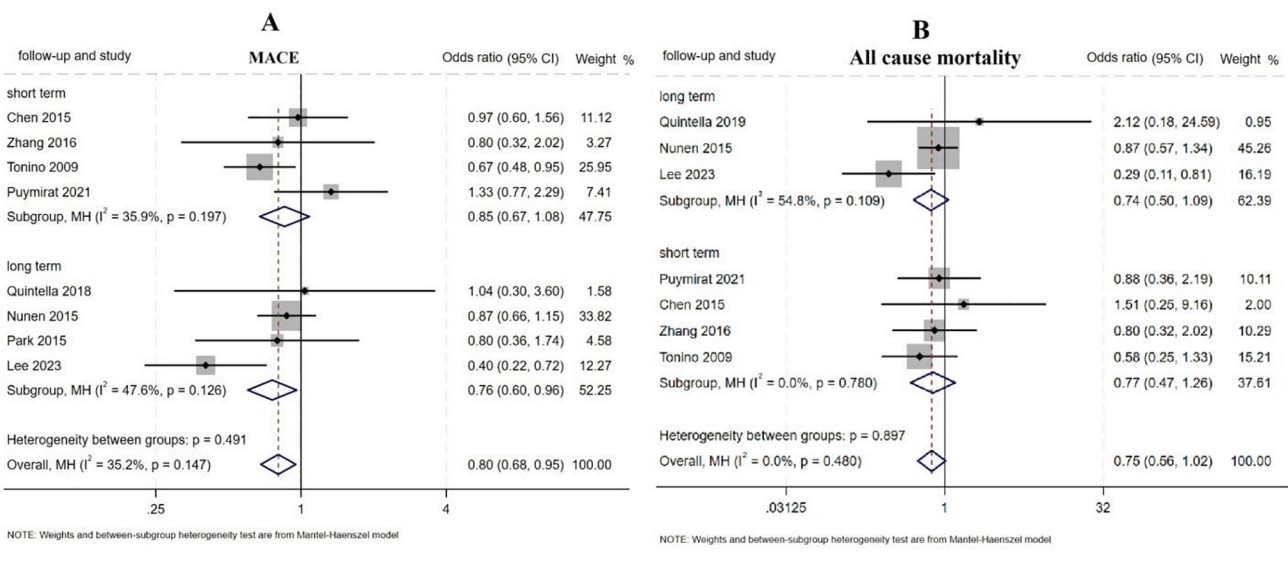

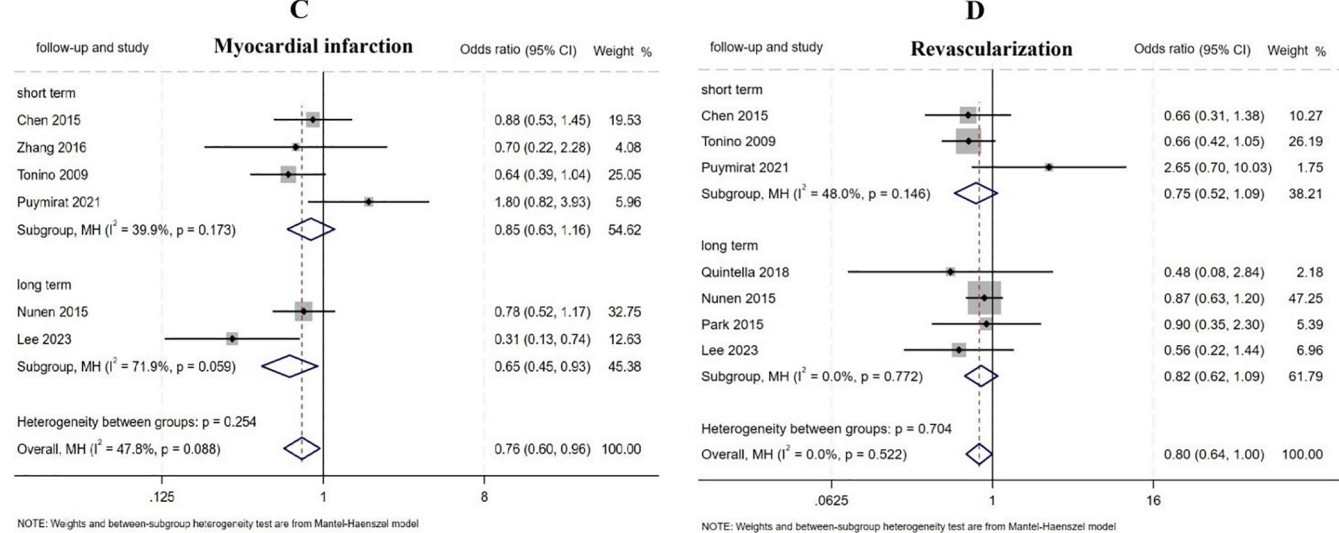

**Fig 2. Forest plot of subgroup analysis comparing FFR-guided versus CAG-guided PCI stratified by follow-up duration.**

3 studies with outcomes beyond 1 year [14,15,17]. Stratified analysis by follow-up duration revealed no significant differences in either the short-term (OR: 0.77, 95% CI: 0.47-1.26, P = 0.296, $I^2$ = 0%) or long-term groups (OR: 0.74, 95% CI: 0.50-1.09, P = 0.123, $I^2$ = 54.8%) (Fig 2B). Further stratification by patient cohorts showed that 4 studies focused on non-ACS patients [6,12,14,15], while 3 studies included ACS patients [11,13,17]. Subgroup analysis demonstrated no significant differences in all-cause mortality between FFR-guided and CAG-guided PCI, both in the ACS subgroup (OR: 0.60, 95% CI: 0.35-1.02, P = 0.060, $I^2$ = 32.8%) and the non-ACS subgroup (OR: 0.84, 95% CI: 0.58-1.21, P = 0.349, $I^2$ = 0%) (Fig 3B). Sensitivity analysis confirmed the consistency of these findings, as the pooled effect sizes calculated using both fixed-effect and random-effects models remained stable across subgroups.

**MI.** Regarding myocardial infarction, 6 studies involving 4135 patients reported MI incidence [6,11–13,15,17], including 4 studies with 1-year outcomes [6,11–13] and 2 studies with outcomes beyond 1 year [15,17]. Among these, 3 studies focused on non-ACS patients [6,12, 15], while 3 included ACS patients [11,13,17]. Subgroup analysis revealed that FFR-guided PCI was associated with significantly fewer MI events compared to CAG-guided PCI in both the long-term group (OR: 0.65, 95% CI: 0.45-0.93, P = 0.018, $I^2$ = 71.9%) (Fig 2C).and the non-ACS subgroup (OR: 0.76, 95% CI: 0.58-0.99, P = 0.039, $I^2$ = 0%) (Fig 3C). However, no significant differences were observed in the short-term group (OR: 0.85, 95% CI: 0.63-1.16, P = 0.307, $I^2$ = 39.9%) or the ACS subgroup (OR: 0.77, 95% CI: 0.47-1.25, P = 0.294, $I^2$= 77.1%). The high $I^2$ value in the ACS subgroup suggested substantial heterogeneity. Upon reviewing the included studies, significant clinical heterogeneity was identified. First, the patient populations differed: Puymirat et al. [11] exclusively enrolled ST-segment elevation MI (STEMI) patients, Zhang et al. [13] included non-ST-segment elevation MI (NSTEMI) and elderly Chinese patients, while Lee et al. [17] enrolled both STEMI and NSTEMI patients. Second, the follow-up durations varied: Puymirat et al. [11] and Zhang et al. [13] followed patients for 1 year, whereas Lee et al. [17] extended follow-up to approximately 3.5 years.

**TVR.** TVR was reported in 7 studies involving 4213 participants. Subgroup analysis based on follow-up duration revealed no significant differences in TVR rates between FFR-guided and CAG-guided PCI, both in the short-term (OR: 0.75, 95% CI: 0.52-1.09, P = 0.196, $I^2$ = 48%) and long-term subgroups (OR: 0.82, 95% CI: 0.62-1.09, P = 0.243, $I^2$ = 0%) (Fig 2D). However, FFR-guided PCI was associated with significantly fewer revascularizations in the non-ACS subgroup (OR: 0.78, 95% CI: 0.61-0.99, P = 0.036, $I^2$ = 0%). In contrast, no significant difference was observed in the ACS subgroup (OR: 0.98, 95% CI: 0.48-2.02, P = 0.315, $I^2$ = 71.3%) (Fig 3D). Sensitivity analysis confirmed the stability and robustness of these outcomes.

## Discussion

This study systematically compared the clinical outcomes of FFR-guided PCI versus CAG-guided PCI in terms of major adverse cardiovascular events, all-cause mortality, myocardial infarction , and target vessel revascularization through meta-analysis and stratified analysis. Overall, FFR-guided PCI demonstrated significant benefits in reducing MACE and MI rates, particularly in non-ACS patients and long-term follow-up, while no significant differences were observed in all-cause mortality. Subgroup analyses further highlighted the advantages of FFR-guided PCI in non-ACS patients.

FFR-guided PCI significantly reduced MACE rates in the long-term follow-up, but no significant difference was observed in the short-term follow-up. Subgroup analysis revealed that FFR-guided PCI significantly lowered MACE rates in non-ACS patients, while no significant difference was observed in ACS patients. A previous large-scale meta-analysis by Zhang.et al.encompassing 49, 517 patients with non-ACS further corroborates the advantages of FFR-guided PCI,which demonstrated that FFR-guided PCI was associated with significantly lower MACE/MACCE compared with CAG-guided PCI strategy [8].

In both short- and long-term follow-up, no significant differences in all-cause mortality were noted between FFR-guided and CAG-guided PCI, with this consistency extending to both ACS and non-ACS subgroups. In contrast, FFR-guided PCI conferred a significant long-term benefit in reducing myocardial infarction (MI) rates, though no such advantage was evident in the short term. This risk reduction was particularly prominent in non-ACS patients, with no significant effect observed in the ACS subgroup. For target vessel revascularization (TVR), FFR guidance significantly lowered event rates in non-ACS patients but failed

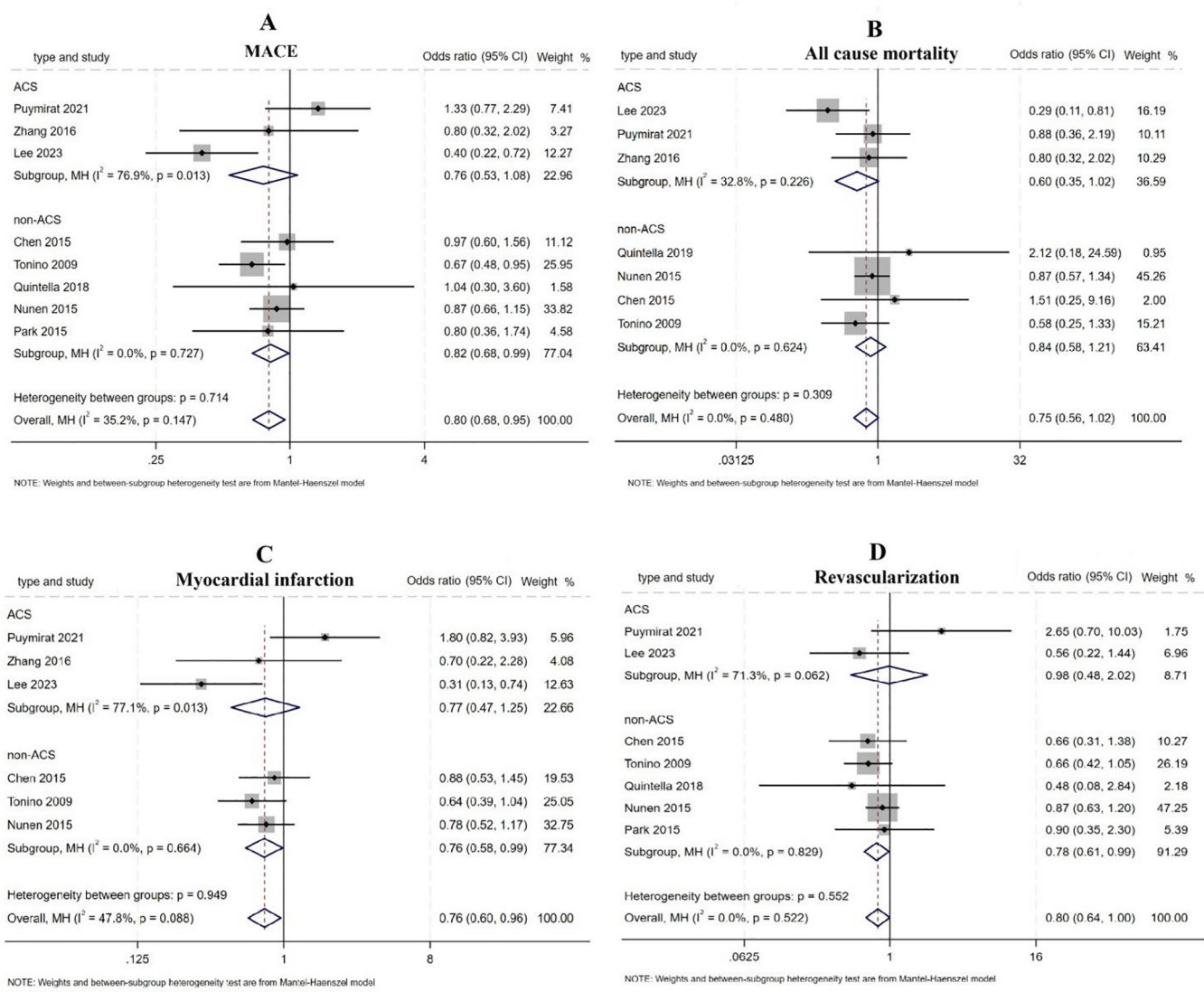

**Fig 3. Forest plot of subgroup analysis comparing FFR-guided versus CAG-guided PCI stratified by patient cohorts.**

to demonstrate a meaningful benefit in ACS patients. Notably, across the overall cohort, FFR-guided PCI did not yield a significant reduction in TVR rates during either short- or long-term follow-up. Consequently, subgroup analysis revealed that in ACS patients, FFR-guided PCI does not outperform CAG-guided PCI across all endpoints—including MACE, all-cause mortality, MI, and TVR. In contrast, it appears to confer benefits in non-ACS patients, with such superiority only becoming apparent in long-term follow-up.

A meta-analysis by Enezate T et al. [18], encompassing 51,350 ACS and non-ACS patients, demonstrated that FFR use was associated with a significantly lower likelihood of MI, MACE, and TLR, though no such association was observed for all-cause mortality. The overall superiority of FFR may largely stem from the overrepresentation of non-ACS patients in the study (51,174/51,350, 99.7%)—a finding consistent with our hypothesis that FFR-guided PCI

confers benefits specifically in non-ACS populations. Another study by Omar A et al. compared CAG-guided versus FFR-guided complete revascularization in patients with STEMI and MVD [19]. Notably, this study revealed that CAG-guided complete revascularization was associated with a significantly reduced risk of MI, all-cause death, and cardiovascular death compared with the FFR-guided approach.This finding further supports our hypothesis that the FFR-guided strategy is not superior to CAG-guided PCI in ACS patients.

The advantages of FFR-guided PCI in non-ACS patients likely stem from the stable pathophysiology of chronic coronary syndromes, enabling precise identification of ischemia-causing lesions and optimized revascularization strategies. By avoiding unnecessary interventions in non-ischemic lesions, FFR-guided PCI reduces procedural complications and late adverse events, such as stent thrombosis or restenosis, thereby improving long-term outcomes.Secondly, the complex and dynamic nature of ACS—including plaque instability, thrombus formation, a prothrombotic state, and microvascular dysfunction—may limit the effectiveness of FFR-guided PCI. Physiological assessment in ACS can be confounded by these acute changes, resulting in less reliable measurements and suboptimal revascularization decisions. The reason why the FFR-guided strategy confers greater clinical benefits compared with the CAG-guided approach is that the absence of significant short-term benefits likely indicates that clinical advantages require time to manifest—with early outcomes being more strongly influenced by procedural factors than by physiological assessment.

Consequently, FFR should be avoided in patients presenting with ACS, even though complete revascularization is associated with fewer clinical adverse events compared with culprit-only revascularization guided by FFR or CAG. Given that functional assessment is limited by acute changes in the ACS state, intracoronary imaging (e.g., OCT or IVUS) should be considered when performing complete revascularization in patients presenting with ACS.

## Innovations and limitations

The innovations of the study include a multidimensional stratified analysis by follow-up duration and patient type, providing comprehensive and precise clinical evidence, as well as comprehensive sensitivity analysis further verified the stability of the results, enhancing their credibility. However, several limitations should be noted. First, a significant gender imbalance was observed, with male patients predominating (3437 vs. 996), likely due to the higher prevalence of coronary heart disease in males, potentially introducing bias and limiting the generalizability of the findings. Second, due to the limited number of included studies (fewer than ten), funnel plot analysis for publication bias was not performed, as the test power would be insufficient to detect potential asymmetry. Additionally, to avoid duplication, two studies from the FAME trial (Nunen 2015 and Tonino 2009), reporting outcomes at different follow-up times, were analyzed separately through stratified analysis rather than pooled together. Lastly, clinical outcomes can be affected by multiple factors, including treatment regimens, smoking history, patient habits, and comorbidities. Addressing these limitations in future research will further strengthen the evidence base for FFR-guided PCI.

## Conclusions

This meta-analysis demonstrates that FFR-guided PCI significantly reduces MACE and MI rates; however these benefits are confined to non-ACS patients and evident only in long-term follow-up. The FFR-guided strategy was not associated with fewer clinical adverse events in patients with ACS compared with CAG-guided approach. Consequently, FFR should be avoided in patients presenting with ACS in clinical practice.

## Supporting information

**S1 Fig. Plots of risk of bias assessment for each item.**
(PNG)

**S2 Fig. Traffic light plots for assessing the risk of bias.**
(PNG)

**S1 File. PRISMA 2020 checklist.**
(DOCX)

**S1 Appendix. Sensitivity analysis of comparing different effect models.**
(PNG)

**S1 Table. Table of all studies.**
(XLSX)

**S2 Table. Table of search items,filters and results.**
(XLSX)

**S3 Table. Table of risk of bias assessment.**
(XLSX)

## Author contributions

**Conceptualization:** Fahui Yin, Yong Zhang, Yangang Chen.

**Data curation:** Xueqian Zhang, Yangang Chen.

**Formal analysis:** Fahui Yin.

**Methodology:** Fahui Yin.

**Resources:** Fahui Yin, Xueqian Zhang, Xuelian Cui.

**Software:** Fahui Yin, Yangang Chen.

**Supervision:** Fahui Yin, Yong Zhang, Xuelian Cui.

**Validation:** Fahui Yin.

**Writing – original draft:** Fahui Yin.

**Writing – review & editing:** Fahui Yin.

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
