## [Decision Letter · Decision Letter 0]

4 Nov 2024

PONE-D-24-28817Updated Mata-Analysis of Fractional Flow Reserve versus Coronary Angiography for Guiding Percutaneous Coronary Intervention.PLOS ONE

Dear Dr. yin,

Thank you for submitting your manuscript to PLOS ONE. After careful consideration, we feel that it has merit but does not fully meet PLOS ONE’s publication criteria as it currently stands. The topic of this meta analysis is clinically important. However, our reviewers pointed out some loopholes and limitations in the methods, and recommended a more comprehensive review of some existing studies. Therefore, we invite you to submit a revised version of the manuscript that addresses the points raised during the review process.

We look forward to receiving your revised manuscript.

Kind regards,

Haipeng Liu

Academic Editor

PLOS ONE

3. As required by our policy on Data Availability, please ensure your manuscript or supplementary information includes the following:

Additional Editor Comments:

Overall, the topic of this meta analysis is clinically important. However, our reviewers pointed out some loopholes and limitations in the methods, and recommended a more comprehensive review of some existing studies. Please carefully revise according to their comments.

Reviewers' comments:

Reviewer's Responses to Questions

**Comments to the Author**

1. Is the manuscript technically sound, and do the data support the conclusions?

Reviewer #1: No

Reviewer #2: Partly

Reviewer #3: Partly

2. Has the statistical analysis been performed appropriately and rigorously? 

Reviewer #1: No

Reviewer #2: No

Reviewer #3: No

3. Have the authors made all data underlying the findings in their manuscript fully available?

Reviewer #1: Yes

Reviewer #2: Yes

Reviewer #3: Yes

4. Is the manuscript presented in an intelligible fashion and written in standard English?

Reviewer #1: No

Reviewer #2: Yes

Reviewer #3: No

5. Review Comments to the Author

Reviewer #1: In this study, Yin et al reported on outcomes of FFR-guided PCI vs Angiography-guided PCI as an updated meta-analysis incorporating recent results from FLOWER MI and FRAME AMI . I have the following comments:

1) Abbreviations should be explained in the main body of the text when they first appear.

2) The DEFER study is the first FFR vs angio-guided PCI study (not FAME) and should be included in the analysis. https://pubmed.ncbi.nlm.nih.gov/11413082/ AND https://pubmed.ncbi.nlm.nih.gov/17531660/

3) This study compared FFR vs angio-guided PCI in NSTEMI and should be included in the analysis.

https://www.ncbi.nlm.nih.gov/pmc/articles/PMC8236005/

4) FLOWER MI has 1 year MACE data and should be included in the analysis

5) Why is the main study for FRAME AMI not quoted? Instead a subgroup analysis from a conference abstract was quoted in its place. Please use the FRAME AMI main study data and reference: https://pubmed.ncbi.nlm.nih.gov/36540034/

6) In your first results paragraph comparing MACE rates of FFR-guided PCI vs angio-guided PCI, you stated that FFR PCI was associated with significant reduction of MACE during short term and long terms, with both 95% confidence intervals appropriately below 1.00, but the p values are non-significant (0.476 and 0.126)- is this an error?

7) You should consider doing a subgroup analysis of ACS vs non ACS RCTs

8) Is the meta analysis done with fixed or random effects? Please state

9) Please consider doing a meta-regression analysis; Is there evidence of publication bias?

10) There are many spelling mistakes- please check the manuscript thoroughly for spelling errors: Eg: Your title should be "meta-analysis" not "mata-analysis"

Reviewer #2: This manuscript presents an updated meta-analysis comparing Fractional Flow Reserve (FFR) and Coronary Angiography (CAG) for guiding Percutaneous Coronary Intervention (PCI). It evaluates key clinical outcomes, including Major Adverse Cardiovascular Events (MACE), all-cause mortality, myocardial infarction (MI), and target vessel revascularization, with follow-up periods divided into short-term (≤1 year) and long-term (>1 year). The study concludes that FFR-guided PCI is associated with a lower incidence of MACE in both short and long-term follow-up, though no significant differences are observed for other outcomes compared to CAG-guided PCI.

1. Gender Bias:

With the majority of patients being male, the authors should explore the potential impact of this imbalance on generalizability and explicitly acknowledge it in the limitations.

2. Comparison with Existing Meta-Analyses:

Including a more detailed discussion comparing the current findings with similar meta-analyses would provide important context and help highlight the novelty of the study’s conclusions.

3. In-depth discussion on pathophysiological factors:

Many pathophysiological factors can influence the treatment efficacy of PCA. Especially, microvascular dysfunction can significantly influence the treatment efficacy and prognosis but is challenging to evaluate (10.1631/jzus.B2100425). An in-depth discussion on other pathophysiological factors is expected.

4. Improve Statistical Rigor and Reporting:

a. Clarify Model Selection: The manuscript should specify when and why a particular model (e.g., fixed-effect or random-effect) was chosen. A fixed-effect model assumes that all studies estimate the same effect size, which works well if heterogeneity is low (I² < 50%). In contrast, a random-effect model accounts for between-study variability and is more appropriate when heterogeneity is substantial (I² > 50%). For example, the use of a random-effect model for long-term MI outcomes (I² = 71.8%) should be justified to improve transparency.

b. Interpret Sensitivity Analyses: Although the manuscript mentions a leave-one-out sensitivity analysis, further explanation is needed on why certain studies (e.g., Nunen et al.) introduced heterogeneity and how their exclusion influenced the overall results. This would provide deeper insights into the robustness of the findings and clarify the sources of variation.

c. Assess Publication Bias: The manuscript does not address potential publication bias, which can distort meta-analytic results. Including funnel plots and performing statistical tests (e.g., Egger’s regression test) would improve rigor and ensure that the findings are not skewed by selective reporting.

Reviewer #3: Overall comments:

This review is well-structured and answers an important research question. It needs careful proofing to improve quality of grammar and spelling to be fit for a journal.

There are other overall concerns such as a lack of clear protocol, a half-completed PRISMA checklist, and statements regarding ethics or conflict of interests, leaving much to be desired to satisfy a reader regarding methodological quality of this work in its current form.

Abstract:

Line 1: The title of the review is misspelled. Please change from “mata-analysis”

Line 8: What does “were medical searched” refer to?

Line 14-17: Check spacing and typography when specifying results of meta-analysis. 95% CI, P-values etc. should be written appropriately to avoid confusion.

Line 21: short and long term follow-up seems to be repeated

Introduction:

Lines 32-44: It is important to clearly explain the research gap and why you expect this gap to exist. From what has been written it is not clear why FFR may be superior to angiography, and if guidelines are clear on the gold standard approach what this review will add

Lines 50-55: Just because other reviews did not compare RCTs does not mean their results are not significant, as evidenced by guidelines agreeing with them and recommending FFR. When doing a redo meta-analysis it is important to justify what will be different to the previous ones, and why there is still a research need.

Methods:

Line 60: Was this protocol conducted in line with PRISMA statement? Was there a pre-specified protocol published on a repository such as PROSPERO? It is important to specify whether such methodological standards were met.

Line 63: Mention whether the search strategy is available to the reader

Line 67: This is vague, were any specific types of patients excluded based on comorbidities or indication for PCI?

Line 68: Does this mean you will aggregate results for all types of patients as long as they all receive PCI? Will this reduce generalisability to your results and did you plan to do a subgroup analysis?

Line 74-75: Why is short-term 1 year and beyond this long-term? In the context of PCI and intervention, typically 5 and 10 year results are thought to be long-term.

Line 83: Please clarify which clinical outcomes and baseline characteristics were extracted

Line 84: Why RoB and not RoB2?

Line 88: PRISMA refers not only to statistical analysis, move this to the start of methods.

Line 89-95: This section describes what you did in the meta-analysis but fails to adequately describe anything else regarding your work. How did you process and present the data for your baseline characteristics? How did you determine which variables would be meta-analysed? How did you determine whether heterogeneity was significant? How did you assess for publication bias or quality of evidence? These questions and the overall purpose of the analysis are not clear from what you have written.

Results:

Line 122: What does the range (14.2-36.4%) with diabetes indicate and how was this calculated? Please clarify this in the methods and update results section of manuscript to match.

Line 124: This is the first mention of the protocol, what is this and where can it be found? Please elaborate in results

Line 125: The length of follow-up is not consistent, will results be aggregated across multiple studies with varying length of follow-up? Please report what this average follow-up length is for the reader also.

Line 148: Please mention how many patients were included in each analysis. This would benefit reader for all meta-analysis conducted.

Line 149: PCI has a lower rate of MACE but the p-value is 0.476? This is not significant please verify the number is correct across all meta-analysis.

Line 178: What was the significance of this heterogeneity and how was it determined?

Discussion:

Line 199: Why would you refer to a trial included within your meta-analysis to justify your results are in line with previously established results?

Line 207-208: It would be pertinent to explain how MACE is reduced in FFR when mortality, TVR, and MI were the same in both groups, both clinically and within your study results.

Line 209: But you only included the FAME study at 1-year of follow-up? When there was longer data available? What was the reason for this and mention in methods how you approached trials reporting multiple sets of results (i.e. from different time periods or subgroups).

Line 210: What was the result of this meta-analysis and how was your study similar/different? Compare and contrast number of patients and meta-analysis results more thoroughly.

Line 214: DEFER-DES is included within your meta-analysis, of course your results will match what they did. Please explain why your result was why it was the case and how conducting your meta-analysis added to the result.

Line 219: It is not appropriate to use rhetoric such as “Definitely not!” in a manuscript.

Line 220: Is this not what guidelines have been saying already based on previous analysis? How does your meta-analysis contribute towards existing understanding of FFR and its usage in clinical practice?

Line 222: This section of the manuscript would benefit from significant addition, as it is unclear as to why FFR is superior to CAG clinically from what has been written thus far.

Line 228: Having all the patients receive PCI does not reduce confounding; it simply makes them appropriate to compare since they are all undergoing the same intervention.

Line 229-230: But the 1-year cutoff was used arbitrarily and is not in line with previous literature (see my comment from earlier)

Line 231: This is a limitation of the inclusion criteria which can be rectified – limitations of the study ideally are things out of your control.

Line 234: Has this been justified before? If so, please reference.

Line 236: This manuscript says “thirdly” abruptly. Please complete this section.

Conclusions:

Line 240: Expand on the context of the result and how this study aids practice

6. PLOS authors have the option to publish the peer review history of their article (what does this mean?). If published, this will include your full peer review and any attached files.

Reviewer #1: No

Reviewer #2: No

Reviewer #3: **Yes: **Niraj S Kumar

---

## [Author Response · Author response to Decision Letter 1]

20 Dec 2024

Dear editors and reviewers:

Tank you for your letter and the reviewers’ comments concerning our manuscript entitled “Updated Meta-Analysis of Fractional Flow Reserve versus Coronary Angiography for Guiding Percutaneous Coronary Intervention”. These comments are very important and highly valuable for me to revise the manuscript and improve the quality of the meta analysis. We have read these comments carefully and modified related issues. We hope the revised version of manuscript could meet the standard of the journal and the approve of the reviewers. The revised portion were marked in red in the manuscript. The point to point responds to the comments of the editors and three reviewers were as following:

Academic editor:

Comment 1: 1. Please ensure that your manuscript meets PLOS ONE's style requirements, including those for file naming. The PLOS ONE style templates can be found at:

Response 1: According to the requirements of PLOS ONE, we throughly checked up the manuscript and modified those which did not fulfill the requirements of the journal.

2.Comment 2: Please include captions for your Supporting Information files at the end of your manuscript, and update any in-text citations to match accordingly. Please see our Supporting Information guidelines for more information:

Response 2: According to your advice, we included captions for Supporting Information files at the end of the manuscript.

Comments 3(a): As required by our policy on Data Availability, please ensure your manuscript or supplementary information includes the following:

Response3(a): According to your advice, we uploaded the numbered table of included and excluded studies identified in the literature search.

Comments 3(b): For every excluded study, the table should list the reason(s) for exclusion.

Response3(b): According to your advice, the reasons for exclusion were included in the table.

Comments 3(c):If any of the included studies are unpublished, include a link (URL) to the primary source or detailed information about how the content can be accessed.

Response3(c): All the included studies have been published, and the full text have been obtained.

Comments 3(d):A table of all data extracted from the primary research sources for the systematic review and/or meta-analysis. The table must include the following information for each study:

Response3(d): A table of all data extracted from the primary research sources for the systematic review were uploaded as supplement materials, and the name of data extractors and date of data extraction were also included.

Reviewer#1

Comment 1: Abbreviations should be explained in the main body of the text when they first appear.

Response 1: We are very sorry for our negligence of the abbreviations. According to your comments, we made correction of the manuscript and checked the full body of the paper.

Comment 2: The DEFER study is the first FFR vs angio-guided PCI study (not FAME) and should be included in the analysis.

Response 2: Indeed, the DEFER study is the first randomised controlled study comparing FFR-guided and angiography-guided PCI. which was published in 2001, all the participants enrolled the study were undergone coronary balloon dilatation. However, the outcomes of the the DEFER study were event-free survival and the percentage of patients free from angina which were not our predefined outcomes. The DEFER-DES Trial was a prospective, randomized study conducted in 6 university hospitals in Korea, and the later one satisfy our inclusion criteria.

Comment 3: This study compared FFR vs angio-guided PCI in NSTEMI and should be included in the analysis.

Response 3: We carefully read the article entitled“Total versus staged versus functional revascularization in NSTEACS patients with multivessel disease” recommended by the reviewer, This RCT was a three-armed trail comparing total revascularization group (total group), staged revascularization group (staged group), and functional revascularization group of NSTEACS patients. Which were different from our enrolled two-armed RCT studies. Perhaps the pooled analysis of two and three armed RCT do exist in previous meta-analysis, however, this kind of selective meta analysis seemed inappropriate, and probably bring certain bias to the overall results of the meta analysis.

Comment 4: FLOWER MI has 1 year MACE data and should be included in the analysis.

Response 4: According to your suggestion, we carefully read the article and extracted the data related MACE. Finally we remade the meta analysis.

Comment 5: Why is the main study for FRAME AMI not quoted? Instead a subgroup analysis from a conference abstract was quoted in its place. Please use the FRAME AMI main study data and reference: https://pubmed.ncbi.nlm.nih.gov/36540034/

Response 5: I sincerely express the great appreciation to the reviewer #1 for providing me the publication version of FRAME AMI study which I searched for a lot of time. The main study of FRAME AMI would great improve the quality of the present meta-analysis. We have replaced the conference abstract with the main study in reference, the other places such as risk of bias assessment were also made correction in the manuscript.

Comment 6: In your first results paragraph comparing MACE rates of FFR-guided PCI vs angio-guided PCI, you stated that FFR PCI was associated with significant reduction of MACE during short term and long terms, with both 95% confidence intervals appropriately below 1.00, but the p values are non-significant (0.476 and 0.126)- is this an error?

Response 6: When we perform the meta analysis with the software of stata 18.0 version, the outcomes of P value usually is responsible for the heterogeneity test. When I2 is great than 50%,the value of P will be less than 0.05,which can been found in the forest plot ; when I2 less than 50%, the P value will become greater than 0.05. In the revised manuscript, we change the P value of the each outcomes, which is responsible for the total combined effect size.However, this value cannot been found in forest plot.

Comment 7: You should consider doing a subgroup analysis of ACS vs non ACS RCTs.

Respomse 7: According to your advice, we performed a subgroup analysis based on patients categories (ACS vs non-ACS) .

Comment 8: Is the meta analysis done with fixed or random effects? Please state.

Response 8: Owing to the less value of I2, no more than 50%, the effects model was used fixed effects model.In the sensitivity analysis, we perform both fixed and random effects model to compare the total effects of the two models.

Comment 9: Please consider doing a meta-regression analysis; Is there evidence of publication bias?

Comment 9: In this meta analysis, we performed subgroup analysis to detect the origin of the heterogeneity among enrolled study, which is based on follow-up durations and patients categories. Taking consideration of the similarity of the two methods, so the meta-regression analysis was not made in this study. Funnel plot was usually used to detect if there is publication bias existing, however, when the enrolled studies were less than 10, the power of funnel plot was unable to detect the real asymmetry of the enrollled studies. only 8 studies included in our study, so we did not perform the funnel plot, the publication bias probably been existing. We described the above in the limitation section.

Comment 10: There are many spelling mistakes- please check the manuscript thoroughly for spelling errors: Eg: Your title should be "meta-analysis" not "mata-analysis"

Response 10: We are very sorry for the spelling errors, Thank your kind suggestion, we have made the corrections of the spelling mistakes,and carefully read and checked the manuscript before submission.

Reviewer#2

Comment 1: Gender Bias:

With the majority of patients being male, the authors should explore the potential impact of this imbalance on generalizability and explicitly acknowledge it in the limitations.

Response 1: In the limitation section, we point out the gender bias in the meta analysis, and the potential reason might been associated with an higher morbidity rate of coronary heart disease in male patients, and this gender discrepancy could problely produce potential bias to the overall results.

Comment 2: Comparison with Existing Meta-Analyses:

Including a more detailed discussion comparing the current findings with similar meta-analyses would provide important context and help highlight the novelty of the study’s conclusions.

Response 2: In the discussion section, we compared our study with previous published meta-analysis,we did a subgroup analysis which demonstrated that FFR-guided PCI was superior to CAG-guided PCI in long term follow up subgroup and non-ACS subgroup, and no significant difference was observed in ACS group. We also explore the potential reasons from pathophysiologic aspects. This conclusion could not been found in previous published meta analysis and should be the novelty of the present meta analysis.

Comment 3: In-depth discussion on pathophysiological factors:

Many pathophysiological factors can influence the treatment efficacy of PCA. Especially, microvascular dysfunction can significantly influence the treatment efficacy and prognosis but is challenging to evaluate (10.1631/jzus.B2100425). An in-depth discussion on other pathophysiological factors is expected.

Response 3: Indeed, many pathophysiological factors can influence the efficacy of PCI. In this meta analysis we found that FFR-guided PCI was not superior to CAG-guided PCI in ACS subgroup analysis. And try to explore the potential pathophysiologic characteristics in ACS subgroup patients in the discussion section.

Comment 4(a): Improve Statistical Rigor and Reporting:

a. Clarify Model Selection: The manuscript should specify when and why a particular model (e.g., fixed-effect or random-effect) was chosen. A fixed-effect model assumes that all studies estimate the same effect size, which works well if heterogeneity is low (I² < 50%). In contrast, a random-effect model accounts for between-study variability and is more appropriate when heterogeneity is substantial (I² > 50%). For example, the use of a random-effect model for long-term MI outcomes (I² = 71.8%) should be justified to improve transparency.

Response 4(a): Yes, we described the model selection in the statistical analysis section.

Comment 4(b): Interpret Sensitivity Analyses: Although the manuscript mentions a leave-one-out sensitivity analysis, further explanation is needed on why certain studies (e.g., Nunen et al.) introduced heterogeneity and how their exclusion influenced the overall results. This would provide deeper insights into the robustness of the findings and clarify the sources of variation.

Response 4(b): In this revised manuscript we changed the method of sensitivity analysis, by comparing the total effect value of the fixed and random model. As we all known that the fixed effect model usually gives more weight to the large sample study, while random effect model gives more weight to small sample study. By comparing the two model could find if small sample study has an impact on the total effects value.

Comment 4(c): Assess Publication Bias: The manuscript does not address potential publication bias, which can distort meta-analytic results. Including funnel plots and performing statistical tests (e.g., Egger’s regression test) would improve rigor and ensure that the findings are not skewed by selective reporting.

Response 4(c): In the limitation section we described the reason why we did not perform the funnel plot. Because only 8studies enrolled into the meta analysis , and the funnel plot is unable to detect the real asymmetry of the enrollled studies, So the publication bias of the present meta-analysis probably been existing.

Reviewer #3: 

Comment 1: Line 1: The title of the review is misspelled. Please change from “mata-analysis”

Response 1: The mistake have been revised,and we carefully check the full manuscript before re-submission to avoid spelling mistakes.

Comment 2: Line 8: What does “were medical searched” refer to?

Response 2: “Medical search” is an official explanation of searching medical data bases by a set of strict methods, such as predefined search terms, searching methods et al.

Comment 3: Line 14-17: Check spacing and typography when specifying results of meta-analysis. 95% CI, P-values etc. should be written appropriately to avoid confusion.

Response 3: We carefully check the spacing and typography before re-submission.

Comment 4: Line 21: short and long term follow-up seems to be repeated.

Response 4: We have done modifications.

Comment 5: Line 32-44: It is important to clearly explain the research gap and why you expect this gap to exist. From what has been written it is not clear why FFR may be superior to angiography, and if guidelines are clear on the gold standard approach what this review will add.

Response 5: In the introduction section, we described that coronary angiography estimated the coronary stenosis by vision assessment, which usually over-estimates or under-estimates the severity of the intermediate coronary stenosis. FFR could provide an scientific assessment of the target vessel from functional aspect. So it is recommended by many guidelines.the purpose of this meta analysis is to further compare FFR and CAG in guiding PCI.

Comment 6: Line 50-55: Just because other reviews did not compare RCTs does not mean their results are not significant, as evidenced by guidelines agreeing with them and recommending FFR. When doing a redo meta-analysis it is important to justify what will be different to the previous ones, and why there is still a research need.

Response 6: Indeed, the words like “the quality cannot reach high level” seemed arbitrary and inappropriate.we delete the phrase.

Methods:

Comment 7: Line 60: Was this protocol conducted in line with PRISMA statement? Was there a pre-specified protocol published on a repository such as PROSPERO? It is important to specify whether such methodological standards were met.

Response 7: Yes, the protocol was conducted in line with PRISMA statement. A detailed PRISMA statement was attached in the supplement materials. We had ever tried to registered the protocol in official website of PROSPERO, and submitted the materials on the website, however, the website seemed not work well and received on reply. It is really a pity for me and the present study.

Comment 8: Line 63: Mention whether the search strategy is available to the reader

Response 8: Yes, we have the screenshot of the whole process of medical searching. I will upload the materials in the re-submission.

Comment 9: Line 67: This is vague, were any specific types of patients excluded based on comorbidities or indication for PCI?

Response 9: Because the objects of meta analysis are those published studies. So it is difficult to identify if the patients excluded based on comorbidities, however, those patients who had been received CABG excluded from this meta analysis.

Comment 10: Line 68: Does this mean you will aggregate results for all types of patients as long as they all receive PCI? Will this reduce generalisability to your results and did you plan to do a subgroup analysis?

Response 10: No, our inclusion criteria were: (1) Randomised controlled trails (RCTs) ; (2) Comparing FFR-guided and angiography-guided PCI; (3) Reporting at least one of the following outcomes, including MACEs, all-cause mortality, myocardial infarction, and target vessel revascularization. This three requirements must been fully met. Maybe our study inclusion criteria seemed strictly and would exclude many previous published famous studies from our meta analysis, however, with accumulating the evidence of FFR practice, We now could better compare FFR and CAG in specific type of patients, such as receiving PCI patients, meanwhile, we did subgroup analysis based on follow up duration and patients categories which could better compare the two methods under different situations.

Comment 11: Line 74-75: Why is short

---

## [Decision Letter · Decision Letter 1]

23 Jan 2025

PONE-D-24-28817R1Updated Meta-Analysis of Fractional Flow Reserve versus Coronary Angiography for Guiding Percutaneous Coronary Intervention.PLOS ONE

Dear Dr. yin,

Thank you for submitting your manuscript to PLOS ONE. After careful consideration, we feel that it has merit but does not fully meet PLOS ONE’s publication criteria as it currently stands. Therefore, we invite you to submit a revised version of the manuscript that addresses the points raised during the review process.

Please submit your revised manuscript  by Mar 09 2025 11:59PM. If you will need more time than this to complete your revisions, please reply to this message or contact the journal office at plosone@plos.org. Please include the following items when submitting your revised manuscript:

We look forward to receiving your revised manuscript.

Kind regards,

Haipeng Liu

Academic Editor

PLOS ONE

Reviewers' comments:

Reviewer's Responses to Questions

**Comments to the Author**

1. If the authors have adequately addressed your comments raised in a previous round of review and you feel that this manuscript is now acceptable for publication, you may indicate that here to bypass the “Comments to the Author” section, enter your conflict of interest statement in the “Confidential to Editor” section, and submit your "Accept" recommendation.

Reviewer #4: All comments have been addressed

2. Is the manuscript technically sound, and do the data support the conclusions?

Reviewer #4: No

3. Has the statistical analysis been performed appropriately and rigorously? 

Reviewer #4: Yes

4. Have the authors made all data underlying the findings in their manuscript fully available?

Reviewer #4: Yes

5. Is the manuscript presented in an intelligible fashion and written in standard English?

Reviewer #4: Yes

6. Review Comments to the Author

Reviewer #4: The authors discuss that the use of FFR guided PCI compared to angiogrpahy guided PCI. The use of FFR to improve pateint outcomes in chronic coronary syndromes are well established from the previous RCT the authors have included. The use of FFR guided PCI in ACS population is questionable including the results from Flower MI. They are two different populations so agree with the authors separating non ACS vs ACS. However, not sure if this meta-analyses adds to what we already know from the previous RCTs.

One comment I have is that Nunen 2015 and Tonino 2009 are the same patient popuation. If you include both studies into the meta-analysis you will be double counting the FAME study population.

7. PLOS authors have the option to publish the peer review history of their article (what does this mean?). If published, this will include your full peer review and any attached files.

Reviewer #4: No

---

## [Author Response · Author response to Decision Letter 2]

4 Mar 2025

Dear Reviewer#4

Thank you very much for your thoughtful and constructive feedback on our manuscript. We greatly appreciate the time and effort you have dedicated to reviewing our work and providing valuable insights.

Regarding your comment about the potential overlap between the patient populations in Nunen et al. (2015) and Tonino et al. (2009), we fully agree with your observation that both studies derive from the FAME study population. To address this concern and avoid double-counting, we have carefully revised our analysis approach. Specifically, we have refrained from pooling the data from these two studies in the meta-analysis. Instead, we conducted a stratified subgroup analysis to ensure that the FAME study population is not redundantly included. This approach allows us to maintain the integrity of our findings while respecting the methodological considerations you raised.

Furthermore, we have explicitly acknowledged this limitation in the Limitations section of our manuscript, emphasizing our efforts to mitigate potential biases arising from overlapping datasets. We believe this adjustment strengthens the robustness of our analysis and aligns with your recommendation.

Once again, we sincerely appreciate your attention to this important detail. Your feedback has significantly improved the quality of our work. We hope that our revised approach and clarification adequately address your concern. Please do not hesitate to let us know if there are any additional points you would like us to consider.

Thank you for your time and expertise.

Best regards,

Fahui Yin

cardiovascular department� Wuwei City Liangzhou hospital

414380580@qq.com

---

## [Decision Letter · Decision Letter 2]

4 Jun 2025

PONE-D-24-28817R2Updated Meta-Analysis of Fractional Flow Reserve versus Coronary Angiography for Guiding Percutaneous Coronary Intervention.PLOS ONE

Dear Dr. yin,

Thank you for submitting your manuscript to PLOS ONE. After careful consideration, we feel that it has merit but does not fully meet PLOS ONE’s publication criteria as it currently stands. Therefore, we invite you to submit a revised version of the manuscript that addresses the points raised during the review process.

We look forward to receiving your revised manuscript.

Kind regards,

Haipeng Liu

Academic Editor

PLOS ONE

**Journal Requirements:**

Reviewers' comments:

Reviewer's Responses to Questions

**Comments to the Author**

1. If the authors have adequately addressed your comments raised in a previous round of review and you feel that this manuscript is now acceptable for publication, you may indicate that here to bypass the “Comments to the Author” section, enter your conflict of interest statement in the “Confidential to Editor” section, and submit your "Accept" recommendation.

Reviewer #5: All comments have been addressed

Reviewer #6: All comments have been addressed

2. Is the manuscript technically sound, and do the data support the conclusions?

Reviewer #5: Yes

Reviewer #6: Yes

3. Has the statistical analysis been performed appropriately and rigorously? 

Reviewer #5: Yes

Reviewer #6: Yes

4. Have the authors made all data underlying the findings in their manuscript fully available?

Reviewer #5: Yes

Reviewer #6: Yes

5. Is the manuscript presented in an intelligible fashion and written in standard English?

Reviewer #5: Yes

Reviewer #6: Yes

6. Review Comments to the Author

**Reviewer #5: **This meta-analysis provides a valuable update on the comparative efficacy of FFR-guided versus CAG-guided PCI, with rigorous methodology and clinically relevant subgroup analyses. The study aligns with prior evidence while addressing gaps in long-term outcomes and patient-specific effects. The revisions have successfully corrected the typos, enhanced the overall clarity and coherence of the writing, and significantly improved the quality and robustness of the study.

**Reviewer #6:** This is an excellent meta-analysis. Some comments have been provided. Introduction, Results and Discussion sections are wonderful. Method section need minor changes. Well done.

7. PLOS authors have the option to publish the peer review history of their article (what does this mean?). If published, this will include your full peer review and any attached files.

Reviewer #5: **Yes: **Adan Khan

Reviewer #6: **Yes: **Shan Ouyang

---

## [Author Response · Author response to Decision Letter 3]

16 Jun 2025

Dear Editors and Reviewers,

We sincerely appreciate the time and effort you and the reviewers have dedicated to evaluating our manuscript. Your insightful comments have been invaluable in helping us improve the quality of our work. We have carefully considered all suggestions and implemented the necessary revisions as detailed below.

In response to the Academic Editor's concern regarding references, we deeply appreciate your careful review. Upon thorough re-examination, we regretfully identified that references 18, 19, and 20 could not be obtained through standard academic databases. To address this, we have removed these references and carefully revised the relevant sections. We have incorporated the study by Enzate T et al. as the new reference 18, which provides stronger support for our discussion. The content previously citing references 18 and 19 has been replaced by reference 8. All corresponding text in the Discussion section has been meticulously updated to reflect these changes.

Regarding Reviewer #6's valuable suggestions on methodology, we are truly grateful for the thoughtful comments concerning methodological rigor. Following this advice, we have modified the Methods section by adding a comprehensive Risk of Bias assessment, including detailed processes of evaluation and the complete content of ROB2. We believe this addition significantly strengthens the methodological framework of our study.

All modifications have been carefully highlighted in the revised manuscript for your convenience. We sincerely hope these revisions meet with your approval. Finally, we wish to express our deepest gratitude for your guidance throughout this review process. Your expertise has been instrumental in helping us improve our manuscript.

With warm regards and highest esteem,

Fahui Yin

Cardiovascular Department, Liangzhou Hospital

Wuwei City, Gansu Province, China 733000

Email�414380580@qq.com

Tel: +86 13309359278

---

## [Decision Letter · Decision Letter 3]

16 Jul 2025

PONE-D-24-28817R3Updated Meta-Analysis of Fractional Flow Reserve versus Coronary Angiography for Guiding Percutaneous Coronary Intervention.PLOS ONE

Dear Dr. yin,

Thank you for submitting your manuscript to PLOS ONE. After careful consideration, we feel that it has merit but does not fully meet PLOS ONE’s publication criteria as it currently stands. Therefore, we invite you to submit a revised version of the manuscript that addresses the points raised during the review process.

We look forward to receiving your revised manuscript.

Kind regards,

Haipeng Liu

Academic Editor

PLOS ONE

Journal Requirements:

Reviewers' comments:

Reviewer's Responses to Questions

**Comments to the Author**

1. If the authors have adequately addressed your comments raised in a previous round of review and you feel that this manuscript is now acceptable for publication, you may indicate that here to bypass the “Comments to the Author” section, enter your conflict of interest statement in the “Confidential to Editor” section, and submit your "Accept" recommendation.

Reviewer #6: All comments have been addressed

Reviewer #7: (No Response)

Reviewer #8: (No Response)

2. Is the manuscript technically sound, and do the data support the conclusions?

Reviewer #6: Yes

Reviewer #7: Yes

Reviewer #8: Yes

3. Has the statistical analysis been performed appropriately and rigorously? 

Reviewer #6: Yes

Reviewer #7: Yes

Reviewer #8: Yes

4. Have the authors made all data underlying the findings in their manuscript fully available?

Reviewer #6: Yes

Reviewer #7: Yes

Reviewer #8: (No Response)

5. Is the manuscript presented in an intelligible fashion and written in standard English?

Reviewer #6: Yes

Reviewer #7: Yes

Reviewer #8: Yes

6. Review Comments to the Author

Reviewer #6: Overall, this is an excellent Meta-Analysis. Especially the result and discussion section. Only minor recommendation in the manuscript for your information.

Reviewer #7: The authors have conducted a meta-analysis to compare FFR vs. Conventional angio for guiding PCI in ACS and Non-ACS patients., where FFR showed benefit in long term outcomes and in Non ACS patients. While the authors have revised the manuscript incorporating feedback from the original reviewers, there are still some additional points which should be addressed before the manuscript can be considered for publication. These are as follows;

Introduction:

While explaining the rationale for the study, the authors write ‘Consequently, we again compare the clinical outcomes of FFR-guided and CAG-guided PCI through even 61 more rigorous quality control and patients’ selection’ The authors should additionally consider documenting how pooling of randomized data at a large scale would give more clinically relevant findings than non-randomized data.

Methods:

In methods, the authors should make a table for the search strategy used for each database and in the next column, specify the number of results that was obtained after the search string was run. This table should be added in supplementary files and referenced in the main manuscript appropriately. This would ensure reproducibility at a broader level.

Cite it appropriately. ‘Due to the limited number of included studies (fewer than ten), funnel plots were not utilized, as 129 their power to detect asymmetry would be insufficient’

From what I know, it is generally advised to use relative risk as an effect size for meta-analysis when all the included studies are RCTs. The authors have used OR instead. Authors should consider giving an explanation for why OR was used. If they decide to switch to RR instead, it should also be explained while being backed up by some strong references.

Explain more from what authors mean by non-ACS? It’s a vague term and demands more explanation. Does it mean Stable CAD patients?

Results:

It would be better if the results of the ROB 2.0 tool can be visually depicted using the Robvis tool. In supplementary files the authors call the traffic light plots ‘ Review of authors’ judgement’ which, theoretically is not wrong, but is a non-formal way of presentation. The authors should consider changing it to ‘Traffic Light Plots for Risk of Bias Assessment for Randomized Controlled Trials or something like this. Additionally, the current plots are from Rob 2 macros, which is obsolete now. Try making them on Robvis which gives clearer and more appropriate images. Make sure to cite the tool.

For the flow diagram, instead of using the self-made flowchart, trying using PRISMA 2020 version and customize it as per study needs.

Discussion:

In the limitations section, the authors have detailed multiple methodological limitations. The authors should focus more on clinical limitations. Heterogeneity, even if mild for some outcomes, should be mentioned. How pooling ACS and Non-ACS patients together negatively impacts clinical interpretation albeit the subgroup analysis.

Additionally, there are similar meta-analysis published in literature. The authors should discuss their findings in light of these already published meta-analyses. Their questions might be little bit different but they are definitely relevant here. Some I could find are: PMID: 25637372, 36115726

The authors may want to discuss if there may be an effect of underlying medications on the pooled results? Is it possible that some medications that these patients may have been currently on, lead to changes in reported parameters?

Rebuttal Letter:

The authors should give a point-to-point response to each comment by the reviewer, with page and line numbers detailed for each change done. Right now, the authors’ responses are vague, generic and difficult to follow.

Missing Content:

The PRISMA checklist is missing. It’s a basic requirement for any Systematic review or meta-analysis and it is considered incomplete without it, as per the Cochrane guidelines. It should be added to supplementary files.

A graphical abstract would be a nice addition.

Reviewer #8: This is a well-structured and clinically relevant meta-analysis of eight RCTs comparing FFR- versus CAG-guided PCI. The stratified analyses by follow-up duration and patient type (ACS vs non-ACS) add important nuance to the interpretation of FFR’s clinical utility. Given that this is a third-round revision, the manuscript appears to have undergone thoughtful improvements, and no major concerns remain. The core methodology, results, and conclusions are sound. I offer a few final suggestions below—largely aimed at enhancing clarity, polish, and alignment between results and tone of conclusions.

1.The manuscript would benefit from moderate language polishing to improve clarity and readability.

2.For outcomes like MI in ACS patients, heterogeneity was high (e.g., I² = 77%). It would strengthen the discussion to briefly address potential sources (e.g., STEMI vs NSTEMI, timing of FFR during acute phases, variability in clinical endpoints).

3.The authors recommend routine use of FFR-guided PCI across clinical practice. Given the null findings in ACS and short-term follow-up, I would suggest softening the language slightly to better reflect the stratified results. Acknowledging that the strongest benefit is in non-ACS and long-term scenarios would improve alignment with the data.

4.Consider ensuring uniform phrasing for “short-term,” “long-term,” “non-ACS,” etc., throughout the manuscript. Some inconsistencies were noted in hyphenation and formatting.

7. PLOS authors have the option to publish the peer review history of their article (what does this mean?). If published, this will include your full peer review and any attached files.

Reviewer #6: **Yes: **Shan Ouyang

Reviewer #7: **Yes: **Jawad Basit

Reviewer #8: No

---

## [Author Response · Author response to Decision Letter 4]

24 Aug 2025

Dear Respected Reviewers:

We are truly grateful to you for devoting time and expertise to evaluate our work—your detailed comments and insightful suggestions have not only pointed out key areas for optimization but also deepened our understanding of the research’s presentation and academic implications.we have addressed every comment one by one, making targeted revisions to the manuscript (e.g.Revising the plot for the risk of bias assessment using R software with the 'robvis' package, adding a table of search items, filters, and results, polishing the language in the manuscript, and so on.). In the following sections of this letter, we will respond to each point specifically, linking the feedback to the corresponding changes in the revised manuscript for clarity.

For the reviewer #7:

Question 1: In the introduction section,While explaining the rationale for the study, the authors write ‘Consequently, we again compare the clinical outcomes of FFR-guided and CAG-guided PCI through even more rigorous quality control and patients’ selection’ The authors should additionally consider documenting how pooling of randomized data at a large scale would give more clinically relevant findings than non-randomized data.

Response: We sincerely thank you for this key comment. From the perspective of methodology, randomized controlled studies provide more objective and convincing results than non-randomized studies by employing randomization and blinding procedures.However, conducting RCTs in daily practice is more challenging, especially when implementing blinding procedures for patients, investigators, and other relevant personnel. This is the fundamental reason why RCTs are regarded as providing the highest level of evidence in clinical research. In previous meta-analyses or studies regarding the comparison of CAG and FFR in guiding PCI, most were derived from observational studies. With the accumulation of experience and practice, we can now conduct meta-analyses based on high-level RCT evidence.

Question 2: In methods section,the authors should make a table for the search strategy used for each database and in the next column, specify the number of results that was obtained after the search string was run. This table should be added in supplementary files and referenced in the main manuscript appropriately. This would ensure reproducibility at a broader level.

Response: Based on your suggestions, we have created a table including the following columns: database, filter, time, and results. This table has been uploaded as a supplementary file.

Questions 3: From what I know, it is generally advised to use relative risk as an effect size for meta-analysis when all the included studies are RCTs. The authors have used OR instead. Authors should consider giving an explanation for why OR was used. If they decide to switch to RR instead, it should also be explained while being backed up by some strong references.

Response: We appreciate the reviewer’s valuable comment regarding the choice of effect measures. In our meta-analysis, we opted for odds ratios (ORs) rather than relative risks (RRs) for the following reasons:1.ORs are the most widely reported effect measure in cardiovascular intervention studies, particularly in studies comparing FFR- vs. angiography-guided PCI. 2.ORs are mathematically stable in both fixed- and random-effects models, especially when event rates vary across studies. In contrast, RRs can overestimate effects when outcome rates are high (>10%), which might be the case in some PCI trials. 3.While RRs may offer a more straightforward clinical interpretation, ORs are well-established in meta-analyses of binary outcomes, particularly in cardiology research. Importantly, when event rates are low (<10%), ORs approximate RRs, minimizing practical differences in interpretation.

Question 4: Explain more from what authors mean by non-ACS? It’s a vague term and demands more explanation. Does it mean Stable CAD patients?

Response: We appreciate the reviewer's insightful comment regarding the terminology. The term 'non-ACS' in our study specifically refers to patients with stable CAD manifestations, including those with: (1) multivessel disease, (2) intermediate coronary stenosis, and (3) true coronary bifurcation lesions. We have now clarified this definition in the revised Subgroup Analysis section to avoid any ambiguity.

Question 5: Results:It would be better if the results of the ROB 2.0 tool can be visually depicted using the Robvis tool. In supplementary files the authors call the traffic light plots ‘ Review of authors’ judgement’ which, theoretically is not wrong, but is a non-formal way of presentation. The authors should consider changing it to ‘Traffic Light Plots for Risk of Bias Assessment for Randomized Controlled Trials or something like this. Additionally, the current plots are from Rob 2 macros, which is obsolete now. Try making them on Robvis which gives clearer and more appropriate images. Make sure to cite the tool.

Response: According to your advice, we have used R software with packages like ‘robvis’ and ‘ggplot2’ to revise the risk of bias assessment plot. Meanwhile, the name of the plot has been revised (see Supplemental materials).

Question 6: For the flow diagram, instead of using the self-made flowchart, trying using PRISMA 2020 version and customize it as per study needs.

Response: Based on your suggestion, we have revised the flow diagram using PRISMA 2020 version (as described in the manuscript).

Question 7: Discussion:In the limitations section, the authors have detailed multiple methodological limitations. The authors should focus more on clinical limitations. Heterogeneity, even if mild for some outcomes, should be mentioned. How pooling ACS and Non-ACS patients together negatively impacts clinical interpretation albeit the subgroup analysis.Additionally, there are similar meta-analysis published in literature. The authors should discuss their findings in light of these already published meta-analyses. Their questions might be little bit different but they are definitely relevant here. Some I could find are: PMID: 25637372, 36115726.The authors may want to discuss if there may be an effect of underlying medications on the pooled results? Is it possible that some medications that these patients may have been currently on, lead to changes in reported parameters?

Response: As you pointed out, in the limitations section, we initially only listed limitations from a methodological perspective. Following your advice, we have revised this section to include limitations from a clinical standpoint as well. However, certain limitations could not be removed, as these were previously raised by reviewers. Additionally, we have refined the discussion section by incorporating insights from previously published meta-analyses. These suggestions have been extremely helpful. Regarding the impact of medication on clinical outcomes, we have addressed this point in the limitations section.

Question 8: Rebuttal Letter:The authors should give a point-to-point response to each comment by the reviewer, with page and line numbers detailed for each change done. Right now, the authors’ responses are vague, generic and difficult to follow.

Response: Thank you for your suggestion. We have made a point-to-point response to each comment, and each revision has been accompanied by the detailed page and line numbers.

Question 9:The PRISMA checklist is missing. It’s a basic requirement for any Systematic review or meta-analysis and it is considered incomplete without it, as per the Cochrane guidelines. It should be added to supplementary files. A graphical abstract would be a nice addition.

Response: Thank you for your suggestion, actually, we have uploaded the PRISMA checklist in supplement materials(as uploaded in supplement files). we have revised the checklist according to the manuscript.

For the Reviewer #8:

Question 1: The manuscript would benefit from moderate language polishing to improve clarity and readability.

Response: Thank you for your helpful suggestions. We highly value this comment and have promptly acted on it. We have carefully revised the manuscript’s language, including adjusting awkward sentence structures, clarifying vague descriptions, and optimizing the logical connection of content to ensure that our research is presented more clearly. These revisions aim to make the manuscript easier for readers to comprehend, and we believe they have effectively addressed your concern.

Question 2: For outcomes like MI in ACS patients, heterogeneity was high (e.g., I² = 77%). It would strengthen the discussion to briefly address potential sources (e.g., STEMI vs NSTEMI, timing of FFR during acute phases, variability in clinical endpoints).

Response: Tank you for your advice, we have discussed the potential reason of high I2 in the Results section (Page 14 line 183-189).

Question 3: The authors recommend routine use of FFR-guided PCI across clinical practice. Given the null findings in ACS and short-term follow-up, I would suggest softening the language slightly to better reflect the stratified results. Acknowledging that the strongest benefit is in non-ACS and long-term scenarios would improve alignment with the data.

Response: Thank you sincerely for your insightful and constructive suggestions, which have been invaluable in refining the quality of our manuscript. Following your guidance, we have thoroughly revised the content in the Discussion and Conclusion sections. Specifically, we have further emphasized the effectiveness of FFR-guided PCI in patients with non-ACS and the long-term subgroup, clarifying the clinical relevance of these findings as you kindly highlighted. Additionally, we have supplemented our recommendation to avoid the use of FFR in ACS patients, ensuring this key point is presented with greater clarity and alignment with your feedback.

Question 4: Consider ensuring uniform phrasing for “short-term,” “long-term,” “non-ACS,” etc., throughout the manuscript. Some inconsistencies were noted in hyphenation and formatting.

Response: Following your valuable advice, we have conducted a thorough review of these phrases throughout the manuscript, ensuring consistent expression, uniform hyphenation, and standardized formatting.

We sincerely appreciate the reviewer’s thoughtful and constructive comments, which have been invaluable in guiding us to refine our study.

---

## [Decision Letter · Decision Letter 4]

22 Sep 2025

Updated Meta-Analysis of Fractional Flow Reserve versus Coronary Angiography for Guiding Percutaneous Coronary Intervention.

PONE-D-24-28817R4

Dear Dr. Yin,

We’re pleased to inform you that your manuscript has been judged scientifically suitable for publication and will be formally accepted for publication once it meets all outstanding technical requirements.

Kind regards,

Haipeng Liu

Academic Editor

PLOS ONE

Additional Editor Comments (optional):

Reviewer #6:

Reviewer #7:

Reviewers' comments:

Reviewer's Responses to Questions

**Comments to the Author**

1. If the authors have adequately addressed your comments raised in a previous round of review and you feel that this manuscript is now acceptable for publication, you may indicate that here to bypass the “Comments to the Author” section, enter your conflict of interest statement in the “Confidential to Editor” section, and submit your "Accept" recommendation.

Reviewer #6: All comments have been addressed

Reviewer #7: All comments have been addressed

2. Is the manuscript technically sound, and do the data support the conclusions?

Reviewer #6: Yes

Reviewer #7: Yes

3. Has the statistical analysis been performed appropriately and rigorously? 

Reviewer #6: Yes

Reviewer #7: Yes

4. Have the authors made all data underlying the findings in their manuscript fully available?

Reviewer #6: Yes

Reviewer #7: Yes

5. Is the manuscript presented in an intelligible fashion and written in standard English?

Reviewer #6: Yes

Reviewer #7: Yes

6. Review Comments to the Author

Reviewer #6: Overall this is an excellent article of meta-analysis in this area. Especially there are some new findings and deep understanding in the discussion section. Well done

Reviewer #7: The authors have thoroughly addressed all of my previous comments, and I have no further concerns. I commend them for their work.

7. PLOS authors have the option to publish the peer review history of their article (what does this mean?). If published, this will include your full peer review and any attached files.

Reviewer #6: **Yes: **Shan Ouyang

Reviewer #7: **Yes: **Jawad Basit

---

## [Editor Report · Acceptance letter]

PONE-D-24-28817R4

PLOS ONE

Dear Dr. Yin,

I'm pleased to inform you that your manuscript has been deemed suitable for publication in PLOS ONE. Congratulations! Your manuscript is now being handed over to our production team.

Kind regards,

on behalf of

Dr. Haipeng Liu

Academic Editor

PLOS ONE